# Appraising Evidence-Based Mental Health and Psychosocial Support (MHPSS) Guidelines—PART I: A Systematic Review on Methodological Quality Using AGREE-HS

**DOI:** 10.3390/ijerph19053107

**Published:** 2022-03-06

**Authors:** Hans te Brake, Andrea Willems, Charlie Steen, Michel Dückers

**Affiliations:** 1ARQ Centre of Expertise for the Impact of Disasters and Crises, 1112 XE Diemen, The Netherlands; a.willems@arq.org (A.W.); c.steen@arq.org (C.S.); m.duckers@arq.org (M.D.); 2Nivel-Netherlands Institute for Health Services Research, 3513 CR Utrecht, The Netherlands; 3Faculty of Behavioural and Social Sciences, University of Groningen, 9712 TS Groningen, The Netherlands

**Keywords:** guidelines, quality, AGREE-HS, mental health and psychosocial support (MHPSS), assessment, DRR

## Abstract

In 2007, the Inter-Agency Standing Committee (IASC) published its guidelines for mental health and psychosocial support (MHPSS) in emergency situations. This was one of the first sets of MHPSS guidelines, developed during the last decades, to aid policymakers and practitioners in the planning and implementation of disaster mental health risk reduction activities. However, the potential merit of MHPSS guidelines for this purpose is poorly understood. The objective of this study is to review available MHPSS guidelines in disaster settings and assess their methodological quality. MHPSS guidelines, frameworks, manuals and toolkits were selected via a systematic literature review as well as a search in the grey literature. A total of 13 MHPSS guidelines were assessed independently by 3–5 raters using the Appraisal of Guidelines for Research and Evaluation–Health Systems (AGREE-HS) instrument. Guideline quality scores varied substantially, ranging between 21.3 and 67.6 (range 0–100, M = 45.4), with four guidelines scoring above midpoint (50). Overall, guidelines scored highest (on a 1–7 scale) on *topic* (M = 5.3) and *recommendations* (M = 4.2), while *implementability* (M = 2.7) is arguably the area where most of the progress is to be made. Ideally, knowledge derived from scientific research aligns with the receptive contexts of policy and practice where risks are identified and mitigated.

## 1. Introduction

Societies in all areas of the world are confronted by a variety of global and local, natural and human-induced hazards and actual disasters [1,2,3]. These events and circumstances entail a serious threat to the mental health and well-being of populations [4,5,6,7,8]. In order to deal with this threat adequately, it is crucial that two areas of expertise are combined and integrated: disaster risk reduction (DRR) and mental health and psychosocial support (MHPSS).

### 1.1. Merging Disaster Risk Reduction and Mental Health and Psychosocial Support

To anticipate disaster exposure risks adequately, governments and partners on a local, national or international level are challenged to pursue an effective DRR strategy integrating the best available knowledge on MHPSS. In the U.N. Glossary, DRR is described as being “aimed at preventing new and reducing existing disaster risk and managing residual risk, all of which contribute to strengthening resilience and therefore to the achievement of sustainable development” [9]. Moreover, DRR is positioned as “the policy objective of disaster risk management, and its goals and objectives are defined in disaster risk reduction strategies and plans” [9]. 

To ensure that DRR effectively addresses mental health risks in particular—i.e., address relevant risk factors for the development of mental health problems at an early stage and strengthen the capacity to respond and adapt to risks and problems, including the provision of recommended services—it is indispensable to integrate the knowledge base of MHPSS into DRR-thinking. A recent review highlighted that this integration can be improved [10]. 

One part of the MHPSS knowledge base consists of decades of epidemiological research into the mental health impact of disasters and relevant risk and protective factors. Another part is embedded in a diversity of documents, originating from a deliberate prioritization of themes and topics, weighing the available evidence and critical discussions among experts from different disciplines, backgrounds and geographies. There are multiple examples. Approximately 15 years ago, the Inter-Agency Standing Committee (IASC)—an inter-agency forum for coordination, policy development and decision making regarding humanitarian assistance involving key U.N. and non-U.N. humanitarian partners—drafted guidelines for MHPSS in humanitarian crisis settings [11]. These guidelines, together with European initiatives focusing on disasters and major incidents [12,13], are early examples of evidence-based MHPSS guidance, in which multiple paradigms are combined (i.e., clinical, psychosocial/social-environmental, and general health care) and seek to provide aid workers and service providers in emergencies with concrete and comprehensive recommendations [14,15,16]. Such guidelines are normative documents that ideally combine scientific knowledge on what works in responding to the plausible mental health burden of disasters and major crises as well as practical knowledge from professionals and policymakers. 

In that sense, these MHPSS documents reflect the same series of activities that are conditional to ensure that DRR, as an iterative process, contributes to consensus as well as a sense of ownership among its crucial stakeholders. MHPSS guideline development projects will result in knowledge, which is especially valuable to DRR when the development manages to establish an effective exchange between the domains of science, policy and practice. Généreux and colleagues distinguished six steps in focused knowledge development towards valorisation across the three domains, including a feedback loop of “knowledge needs” leading back from policy and practice to science (Figure 1) [17]. 

### 1.2. Study Focus

Our assumption is that MHPSS guideline development, when conducted properly, helps to effectively shape DRR from a mental health perspective. The current study is one half of a diptych in which we aim to contribute to a further integration of MHPSS knowledge into the DRR framework. The objective of this first part is to collect the available guidelines and assess their methodological quality. The second part, described in a separate paper by Van Hoof and colleagues [18], contains a qualitative analysis of the contents of the guidelines. 

The focus on methodological quality is important to understand better to what extent the end users of existing guidelines actually are provided with the best available knowledge. How are the multiple MHPSS guidelines different in terms of methodological quality? To answer this question, we build on the work of the AGREE team, an international team of researchers that has developed several tools to appraise (clinical) guidelines [19]. Existing MHPSS guidelines fall in the category of “Health systems guidance”, defined as guidelines that provide “systematically developed recommendations to manage challenges related to health system governance, financial and delivery arrangements, and the implementation strategies needed to get the right programmes and services to those who need them” [20] (p. 1). In 2018, the AGREE collaboration published a tool to evaluate such guidelines: the Appraisal of Guidelines for Research and Evaluation–Health Systems (AGREE-HS) [21].

The assessment of the methodological quality of available guidelines through the AGREE lens is beneficial in the light of the iterative process described in Figure 1, as it adds clarity about the extent to which available standards have a potential to serve as a foundation for practice-oriented policymaking. Against this background, the aim of this paper is to provide an overview of the methodological quality of the available international MHPSS guidelines. 

## 2. Materials and Methods

### 2.1. The Guideline Search

Two search methods were used to search for relevant documents: scientific literature and grey literature. First, a systematic literature search was conducted in 7 databases: PsycInfo (Ovid), Ovid Medline (about 99% the same as PubMed), Embase (Ovid), Ovid Evidence Based Medicine Reviews (Cochrane Library), Published International Literature On Traumatic Stress (PILOTS), Sociological Abstracts, and Web of Science. The search included terms specifying (1) context: disasters, emergency/humanitarian setting; (2) content: PSS, MHPSS and PFA; and (3) form: manual, framework, review, toolkit, implementation plan (see Appendix A for the complete search strategy used). Publications from 2007 onwards were included (being the publication date of IASC’s MHPSS guidelines). An initial search was conducted in 2019, and an update was performed in June 2021. The resulting articles were screened on title and abstract by two independent researchers (C.S. and H.t.B. in 2019/20 and A.W. and H.t.B. in 2021). Discussions were solved by consensus. Articles were included when the abstract mentioned an international guideline, manual, framework, review, toolkit, or implementation plan concerning MHPSS in disaster response and uses it for analysis, comparison or evaluation. 

To identify guidelines beyond those found via the traditional publishing and distribution channels, an additional search was conducted in the ‘grey literature’ (see, e.g., [22]). We used the results of the E.U. project called OPSIC [23], which was developed with the aim to provide psychosocial crisis managers and mental health professionals with an overview of high quality mental health and psychosocial support programing and interventions in the context of disasters. Within OPSIC, an extensive search was performed, including local documents within E.U. countries to arrive to a comprehensive overview of 282 “guidance documents”. In addition to the OPSIC results, we included documents mentioned on two websites: mhpss.net [24] and ReliefWeb [25]. The Emergency Toolkit database on mhpss.net is a growing global platform. The network functions as an online community of practice for mental health and psychosocial support in challenging humanitarian and development contexts [26]. ReliefWeb is a humanitarian information service provided by the United Nations Office for the Coordination of Humanitarian Affairs (OCHA), which collects information from more than 4000 key sources, including humanitarian agencies at the international and local levels, governments, think-tanks and research institutions, and the media [25].

### 2.2. Guideline Selection (Eligibility Criteria)

Inclusion criteria for the documents found in both the scientific literature and in grey literature (all of the following criteria were to be met):The document is a ‘guideline’ as defined by AGREE-HS (i.e., “Health systems guidance or guideline documents are systematically developed statements to assist with decisions about appropriate options for addressing health system challenges, the implementation of these options, and the monitoring and evaluation of the implementation efforts” [21] (p. iii));The document is focused on disaster/emergency situations;The core topic of the document concerns the organization of MHPSS, not the use of (one-on-one) interventions, thus the guidance is focused on the organizational or (inter)national level;When documents stem from a main ‘overarching guideline’, (i.e., they are spin-offs, such as target group specific translations or practical protocols), only the overarching or most recent version of the document is included;The document is available in the English or Dutch language and can be found online.

### 2.3. Appraisal of Methodological Quality

The methodological quality of each included guideline was assessed using AGREE-HS [21]. The AGREE-HS is derived from AGREE II [27] and is specifically developed for use on non-clinical guidelines. It contains 5 core quality items focused on (1) *topic*; (2) *participants*; (3) *methods*; (4) *recommendations*; and (5) *implementability* (see Box 1). A manual was used providing an overall description of using AGREE-HS, as well as the defining criteria for each core item [21,28]. Using the criteria and manual, the guidelines were rated on each topic using a seven-point scale (from 1 = Strongly Disagree to 7 = Strongly Agree), with higher scores indicating that more parameters of the criteria were met. A total of 5 raters were utilized (H.t.B., F.M., A.W., C.S. and E.G.); all guidelines were rated independently by at least 3 raters, with 2 raters (H.T.B and A.W.) rating all guidelines. The professional background of the raters is public health (A.W., C.S. and F.M.), psychology (E.G.) and methodology, social psychology and occupational psychology (H.t.B.). To arrive at an overall score for the guideline, scoring method 1 as mentioned in the manual was used: scores on all items were summarized and transformed to a percentage of the maximum possible score (taking into account the number of raters). This resulted in a score between 0–100, with a higher score indicating a better methodological quality.

Box 1The 5 core quality items of AGREE-HS used to score guidelines (from [21]).
*Topic*
This item addresses the description of the health system challenge, the causes of the challenge and the priority awarded to it, and relevance of the guidance.
*Participants*
This item addresses the composition of the health systems guidance development team and the management of competing interests and funder influence.
*Methods*
This item addresses the use of systematic methods and transparency in reporting; the use of the best available and up-to-date evidence; the consideration of effectiveness and cost-effectiveness of the potential options; and the weighting of benefits and harms in the guidance document.
*Recommendations*
This item addresses the outcome orientation and comprehensiveness of the guidance; the ethical and equity considerations drawn upon in its development; the details for its operationalization; the sociocultural and political alignment of the guidance; and the updating plan.
*Implementability*
This item addresses the barriers and enablers to implementing the recommendations; the cost and resource considerations in implementing the recommendations; the affordability of implementation and anticipated sustainability of outcomes; the flexibility and transferability of the guidance; and the strategies for disseminating the guidance, monitoring its implementation and evaluating its impact.

## 3. Results

### 3.1. Guideline Search and Selection

This study applied the standards from the Preferred Reporting Items for Systematic Reviews and Meta-Analyses statement (PRISMA [29]; see flow chart in Figure 2 and the literature search strategy in Appendix A). The literature search resulted, after removing the duplicates, in 2532 abstracts of which 66 were included after screening. These 66 publications contained references to 16 unique documents, of which 14 could be retrieved. Screening of the documents described by the OPSIC project and the online Emergency Toolkit on mhpss.net and ReliefWeb resulted in 37 and 39 additional unique documents, respectively. Using the criteria of inclusion, a final selection of 13 guidelines remained for quality assessment [11,12,23,30,31,32,33,34,35,36,37,38,39].

Table 1 lists the resulting 13 guidelines and their references. A total of 7 of the 13 guidelines were identified via scientific databases and registers. The IASC Guidelines 2007 are referenced in most (33 references; 29 of these concern contributions in a Special Issue of the journal Intervention in 2008); all others are mentioned in 8 or less publications. Most of the findings from mhpss.net and ReliefWeb overlap with the inclusions from the scientific literature. Only one additional guideline was found using ReliefWeb (i.e., Red Cross Guidelines 2018). In the OPSIC project [22], 10 guidelines were found, 6 of which overlapping with a scientific mention.

### 3.2. Appraisal of Methodological Quality

The methodological quality of the 13 guidelines is summarized in Table 2. The average AGREE-HS score is 45.4, ranging between 21.3 and 67.6 on a scale from 0 to 100. The highest score is found for the Dutch Guidelines 2014, although its score of 67.6 still allows room for improvement. Table 2 also summarizes the results on each of the five AGREE-items, with the bottom row showing the mean AGREE item score. Overall, the items *topic* and *recommendations* scored the highest (all above midpoint of a 1–7 scale), while *participants*, *methods*, and *implementability* scored below the scale mean. The outcomes for each of the five AGREE-HS items are discussed in more detail below.

#### 3.2.1. Topic

The *topic* of the guidelines is well described with an overall mean score of 5.3 (SD = 1.05), ranging between 3.2 (TENTS Guidelines 2008) and 6.6 (Red Cross guidelines 2018). Thus, most guidelines provide an adequate description of which health system challenge is addressed and why this is relevant. Guidelines scoring low on this item are lacking in their description of their rationale, the context in which guidelines were developed, or the context in which they are intended to be used. The absence of an adequate description of the *topic* makes it hard for end users to determine their usefulness.

#### 3.2.2. Participants

The overall mean score on the description of *participants* (i.e., the composition of the health systems guidance development team) is 3.2 (SD = 0.84) and ranges between 1.8 (PFA Field Operations 2006) and 4.8 (Dutch Guidelines 2014). The majority (eight) of the guidelines score below midpoint on this item because it was unclear what type of experts were involved or there was no multisectoral/multidisciplinary approach. Scores on this item are also often lowered by unclarity on the presence and/or influence of funding agencies.

#### 3.2.3. Methods

The *methods* score 3.2 (SD = 1.34) with a relatively large range between 1.3 (PFA Field Operations 2006) and 5.6 (Dutch Guidelines 2014). We found most guidelines to rely on consensus or ‘best practice’; however, most documents do not describe how this consensus is reached.

#### 3.2.4. Recommendations

Most guidelines offer a clear description of *recommendations*; the mean overall score was 4.2 (SD = 0.98), ranging between 3.0 (Korean Guidelines 2019) and 6.3 (Red Cross Guidelines 2018). Lower scores on this item are mostly due to a lack of ‘alignment with a specific sociocultural and/or political context’. Guidelines with low scores on this item also lack a description of expected outcomes and indicators of the recommendations. Few guidelines explicitly outline a plan for updating the recommendations.

#### 3.2.5. Implementability

The overall score for *implementability* is the lowest of all items with 2.7 (SD = 1.02), ranging between 1.6 (PFA Field Operations 2006) and 4.9 (Red Cross Guidelines 2018). Information on the sustainability of the recommendations and long-term outcomes is lacking, and aspects, such as the needed resources and affordability, are scarcely mentioned. A description of strategies on how to assess the implementation process and the impact of recommendations is also absent in most guidelines.

## 4. Discussion

This study aimed to provide insight into the currently available MHPSS guidelines and their methodological quality in order to determine whether such guidelines can aid policymakers and practitioners in the planning and implementation of disaster mental health risk reduction (DMHRR) activities. After a short overview of our findings, we discuss some notable implications for policy and practice.

### 4.1. Retrieving MHPSS Guidelines

In our analysis, we assessed a total of 13 eligible MHPSS guidelines. These guidelines have in common that they can be seen as normative guidance documents, focusing on the nature, contents and organization of MHPSS for populations exposed to disaster or humanitarian crises. The guidelines contain systematically developed statements to explain key principles and imperatives, and to support end users in their MHPSS policy development and decision making in the case of a disastrous event. The statements also intend to provide options to address health system challenges, the implementation of these options, and the monitoring and evaluation of MHPSS. As explained earlier, the contents of the MHPSS guidelines (i.e., the contents of the aforementioned options themselves) are analysed in an accompanying paper.

A substantial amount (15) of documents could not be found online. This makes them unavailable and unusable for end-users who will not have the time to explore the field extensively in the first place. In addition, we found that it is not always clear how different documents/guidelines relate to each other. Even guidelines developed within the same organization are not always complementary or add value to each other.

About half (seven) of the included guidelines could be found via scientific databases and registers, with the IASC Guidelines 2007 referred to the most, followed by the PFA Field Workers 2011 and mhGap guidelines 2015 [11,33,35]. The Korean Guidelines [30] resulted from the scientific literature search only (i.e., these guidelines are only available via peer-reviewed databases). The remaining included guidelines (6) were found via a ‘grey’ search (i.e., OPSIC, mhpss.net, and/or ReliefWeb). 

### 4.2. Improving the Methodological Quality of MHPSS Guidelines

The assessment shows considerable variation in methodological quality between the guidelines. Using the AGREE-HS format as the criterion to judge the methodological quality of guidelines, given the theoretical score range from 0 to 100, even the highest scoring guidelines in our sample leave ample room for improvement. Four guidelines stand out, based on the overall AGREE-HS score: the Dutch Guidelines 2014 [34], the IASC Guidelines 2007 [11], the Red Cross Guidelines 2018 [31] and OPSIC 2016 [23]; all score above the half-way mark of the 0–100 scale provided by AGREE-HS.

The results on the five AGREE-HS items offer some indication for further improvement. The highest scores are found on *topic* and *recommendations*, which is in line with the results reported by Brouwers et al. [40], who also reported the highest scores on these items. Improvement on *topic* is possible, particularly in the description of rationale and context. As argued by Miller et al. [14], MHPSS guidelines should be clear in their description of the ‘why’ and the ‘how’ of the proposed interventions. Dückers suggested to expand this list with ‘for whom’, ‘by whom’, and ‘when’ to further specify what type of information is desirable to provide optimal practical guidance [16]. 

The description of *participants* as well as *methods* has an average score. Even the Red Cross Guidelines 2018 [31], with a high overall score, scores below the 3.5 midpoint on both items. The majority of guidelines are unclear about the involvement of a (multidisciplinary) set of experts. A multi-sectoral action and coordination in guideline development is necessary because individuals, families and communities in emergency settings have problems and needs that cut across sectoral definitions [41]. Concerning *methods*, there seems to be a trade-off between rigid description of methods and the actual usability of the guidelines. The overall score of *methods* is below average. We acknowledge the possibility that many of the documents, including those that receive a low score, are actually based on more rigidly applied methods, but that these just have not been described accordingly. In any case, a less than optimal description of the used methods does not preclude the relevance of the guideline itself (cf. Tol et al. [42]).

All guidelines with an overall score below 50 have in common that their *implementability* scores are below 3, although this is also the case for (high overall-scoring) OPSIC 2016 [23]. Almost all guidelines state words along the lines of “the recommendations should be translated to specific settings and locations taking into account local context, needs and possibilities”. However, an explanation on how to actually achieve this is often omitted, which leaves something to be desired for practical, immediate use. 

Although *implementability* is an important issue, it needs to be put into perspective. Our methods themselves preclude the inclusion of very specific protocols and practice guidelines. That is, in our selection process, we looked for international and ‘overarching’ guidelines, even though more elaborate expansions, specifically aimed at specific target groups or situations, were found. This introduces a ‘catch-22’ in our venture: in aiming to select generalizable guidelines, we also move to more abstract ones, which by definition are less usable for direct implementation. The question remains: even if we do find the best generic evidence-based guidelines, do they provide support for practical use in the field, given the known differences in context, and therefore the inability to provide recommendations for each of these contexts? 

### 4.3. What to Conclude concerning the Implications for Disaster Risk Reduction?

One could argue that the variation identified in methodological quality in general, and in different scoring domains in particular, is indicative for the inevitable alignment challenge mentioned in the Introduction (visualized in Figure 1). The methodological strengths and points of improvement of the MHPSS guidelines from this study can serve as a starting point for future guideline development combining MHPSS and DRR. The dialogue between science, policy and practice, when effectively organized, will contribute to the clarity of the *topic*, the active representation and engagement of potential *participants* (including end-users, target groups and facilitating actors), the use of *methods* that involve the best available scientific evidence, expert knowledge and practical wisdom, outcome-oriented and comprehensive *recommendations* and a highly probable level of *implementability*. When approached in this manner, efforts to achieve the highest methodological quality level of MHPSS guidelines match perfectly with a process of structured knowledge development and the implementation of activities that can help to reduce disaster-related mental health risks. This applies to the global context, but also to crisis-affected local communities where reliable and meaningful guidance on what to do (and not to do) is invaluable. 

What this study suggests, generally, is that the participatory guideline development processes in itself should be seen as a form of community engagement contributing to disaster preparedness. More specifically, knowledge about the methodological quality of MHPSS guidelines, apart from guideline contents (and the themes and issues identified by Gray et al. [10]), provides an opening to promote the desired integration of MHPSS into DRR further. 

We think this is particularly true for participant involvement. When the representation of partners between and within the domains of science, policy and practice is disproportionate, reasonably incomplete or alternatively biased, it will affect other AGREE quality items, such as *methods* and *implementability*. Although participant engagement is a priori beneficial for guideline support and ownership, it is important to remain critical about ‘group think’ and ‘tunnel vision’ in the selection and prioritization of topics and formulation of recommendations, without making concessions when it comes to the evidence base of preferred over less popular measures and interventions. Knowledge development on behalf of disaster risk reduction will be fatally flawed when politically or culturally sensitive topics or other aspects crucial for the planning and effective delivery of services for vulnerable groups (key target groups in the case of MHPSS) are not included and considered in the guideline development process, from start to finish. In that case, the guidance resulting from the process is likely to reflect existing shortcomings that will probably turn out to be problematic when a disaster actually takes place.

### 4.4. Limitations

Although care has been taken to rule out bias in the selection and screening of guidelines (e.g., by using multiple independent judges), such bias cannot be fully dismissed. Guidelines that are in some way the topic of peer-reviewed publications are included. The grey search was performed to check for additional MHPSS guidelines that are widely used, but not published about. Thus, we increased the comprehensiveness of the search (as recommended in [22]). Although we believe to have used the main sources for finding additional documents (i.e., OPSIC, ReliefWeb and MHPSS.net), it is likely that the list is incomplete. Not only because work is progressing and new guidelines are being developed, but also because we limited our sample to publications in the English and Dutch language. Many locally developed guidelines fall beyond the reach of the current search. Evidently, future updates to the current outcomes are recommended and welcomed.

## 5. Conclusions

As stated on the AGREE website, “The potential benefits of practice guidelines are only as good as the quality of the guidelines themselves” [27]. We examined the methodological quality of 13 internationally developed MHPSS guidelines using AGREE-HS. Four guidelines scored above the half-way mark of the 0–100 scale provided by AGREE-HS. The highest scores were found on *topic* and *recommendations*. The items *participants* and *methods* had an average score and *implementability* had the lowest overall score. The variation in quality we found is indicative of the challenge to combine knowledge from science, policy and practice, especially in formulating an adequate response to acute mental health risks. Ideally, knowledge derived from scientific research should align with the context of policymakers and practitioners as it would help to make informed decisions to identify and mitigate risks (which is relevant for disasters and other emergencies as well as ‘normal’ health care settings). The AGREE score offers a transparent indication of the quality of the guidelines and where improvements can be made. An analysis on the pending issues concerning *topic*, *participants*, *methods*, *recommendations*, and *implementability* can provide an opening to further promote the integration of MHPSS into DRR.

## Figures and Tables

**Figure 1 ijerph-19-03107-f001:**
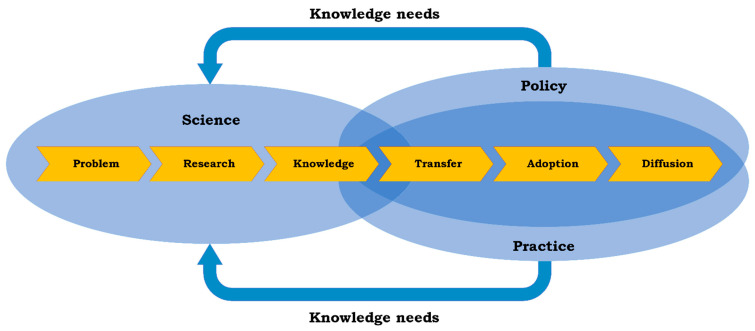
Focused knowledge development and valorisation in three domains (adapted from Généreux et al. [17]).

**Figure 2 ijerph-19-03107-f002:**
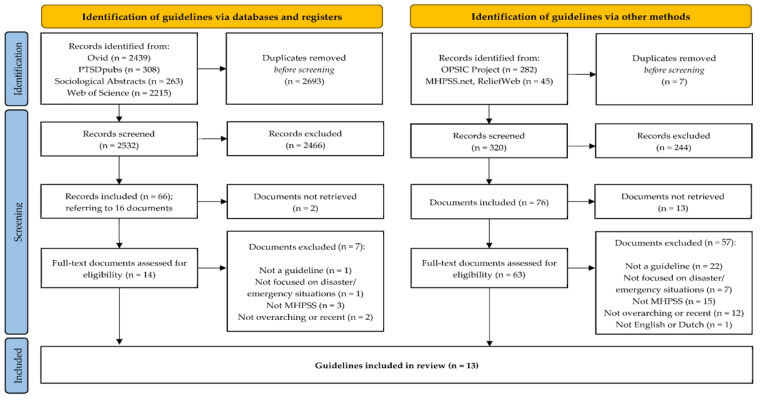
PRISMA diagram of guideline identification and selection (from [29]).

**Table 1 ijerph-19-03107-t001:** Selected MHPSS guidelines and their origin (sorted by year of publication; most recent first).

Origin				Guideline Author(s) [Reference]	Short Title
A	B	C	D		
1	0	0	0	Lee et al. [30]	Korean Guidelines 2019
0	0	1	0	The International Committee of the Red Cross [31]	Red Cross Guidelines 2018
0	0	0	1	European Network for Psychosocial Crisis Management (EUNAD) [32]	EUNAD Guidelines 2017
0	0	0	1	Juen et al. [23]	OPSIC 2016
4	2	1	1	World Health Organization & United Nations High Commissioner for Refugees [33]	MhGAP 2015
2	0	0	1	Werkgroep multidisciplinaire richtlijnontwikkeling psychosociale hulp bij rampen en crises [34]	Dutch Guidelines 2014
6	1	0	1	World Health Organization, War Trauma Foundation & World Vision International [35]	PFA Field Workers 2011
0	0	0	1	DH Emergency Preparedness Division [36]	NHS Guidelines 2009
1	0	0	1	Bisson & Tavakoly [12]	TENTS Guidelines 2008
0	0	0	1	NATO Joint Medical Committee [37]	NATO Guidelines 2008
33	0	1	1	Inter-Agency Standing Committee (IASC) [11]	IASC Guidelines 2007
8	0	0	1	Brymer et al. [38]	PFA Field Operations 2006
0	0	0	1	Seynaeve [39]	European Policy Paper 2001

Origin = type of inclusion: A = scientific literature search; B = mhpss.net (Emergency Toolkit database); C = ReliefWeb; D = OPSIC project. Short title = title assigned by authors for reference.

**Table 2 ijerph-19-03107-t002:** MHPSS guidelines assessment (sorted highest–lowest on total AGREE-HS score).

AGREE (SD)	#Raters	Short Title [Reference]	AGREE Items				
			Topic	Participants	Methods	Recommendations	Implementability
67.6 (8.4)	4	Dutch Guidelines 2014 [34]	6.3	4.8	5.6	4.9	3.8
63.6 (11.7)	4	IASC Guidelines 2007 [11]	6.2	4.7	3.8	5.4	4.1
60.8 (9.6)	4	Red Cross Guidelines 2018 [31]	6.6	2.7	2.8	6.3	4.9
56.8 (11.5)	3	OPSIC 2016 [23]	6.3	3.6	4.5	4.8	2.9
48.5 (2.6)	3	EUNAD Guidelines 2017 [32]	5.8	3.4	4.0	4.3	2.1
44.2 (1.8)	3	NATO Guidelines 2008 [37]	5.3	3.1	3.4	3.7	2.8
41.5 (21.4)	3	Korean Guidelines 2019 [30]	4.5	3.2	4.8	3.0	1.9
41.1 (6.9)	3	European Policy Paper 2001 [39]	5.7	3.0	1.7	3.7	3.3
41.0 (9.6)	3	PFA Field Workers 2011 [35]	5.3	3.7	1.5	4.7	2.1
37.3 (16.0)	4	MhGAP 2015 [33]	5.4	3.0	2.1	3.5	2.3
35.4 (13.1)	3	NHS Guidelines 2009 [36]	5.3	2.3	3.0	3.3	1.8
34.1 (11.4)	5	TENTS Guidelines 2008 [12]	3.2	2.9	3.0	3.3	2.0
31.2 (16.8)	3	PFA Field Operations 2006 [38]	3.5	1.8	1.3	3.3	1.6
		*Mean overall score per AGREE-item:*	*5.3*	*3.2*	*3.2*	*4.2*	*2.7*

AGREE (SD) = total score (range 0–100) calculated following the recommendations given by AGREE-HS Research Team [21] (standard deviation also provided); #raters = total number of raters for this document (all guidelines were scored by at least 3, and maximal 5 raters); short title = title assigned by authors for reference; AGREE items = mean scores (range 1–7) on each of the 5 sub-items of AGREE-HS (*topic*, *participants*, *methods*, *recommendations*, and *implementability*).

## Data Availability

Not applicable.

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
