# Peer review of "Appraising Evidence-Based Mental Health and Psychosocial Support (MHPSS) Guidelines—PART I: A Systematic Review on Methodological Quality Using AGREE-HS"

_ijerph, 2022, doi:10.3390/ijerph19053107_

Round 1

Reviewer 1 Report

Dear Authors,

Thank you for giving me the opportunity to review this interesting and  during my review of the text I had some doubts about the limitations, but these are described in the final section, so I recommend it to publish in present form.

1)The state of the art is well elaborated
2)Research: the research was correctly designed
3)The materials and methods are precisly described, and are appropriate and align with the description of the state-of-the-art and the research objectives
4)Results: the results are well presented and the limitations of research are described
5)Interpretation: the conclusions are reasonable extension of the results

Reviewer 2 Report

Thanks for the opportunity to review the manuscript titled: "Appraising Evidence-Based Mental Health and Psychosocial Support (MHPSS) Guidelines – PART I: A Systematic Review on Methodological Quality using AGREE-HS"

I would like to congratulate the authors for writing this paper. This manuscript is interesting, well-written, organised and the choices made in the research seem to be relevant. 

However, for publications, this paper requires some changes/additions/ and edits, which area minor. I hope these comments and suggestions may be useful for improving your study. Good luck!

- INTRODUCTION: A greater effort is required to identify the contributions derived from this study ad the relevance of the study.

- CONCLUSIONS: Beyond the result discussion, it is necessary to enhance both the theoretical and practical contributions that this article implies to current literature. The authors must make suggestions for the future.

Reviewer 3 Report

The authors provided insight into existing MHPSS guidelines and evaluated their methodological quality to help policymakers and practitioners. The review is robust and compelling. 

Just some minor issues should be addressed: 

  • Please specify what exactly "grey literature" is. The term could be unknown for a multidisciplinary readership.
  • Inter-rater agreement on each of the 5 AGREE-items should be computed and presented. I would suggest using Fleiss' kappa. 
  • In table 2, I think it could be useful to present also the standard deviation of the agreement. Mean values are indeed very sensitive to "extreme values" especially if the number of observations (i.e., raters) is low. 
  • More information about raters is needed to address possible conflicts of interest. Were the raters linked in any way with the guidelines they were evaluating?

Reviewer 4 Report

1) This part of the study is very well described, being a good review of the Mental Health and Psychosocial Support guidelines.
2) Please elaborate on the limitation and implications of not having third country guides not published in English or German.
3) Independent raters are indicated, but they are co-authors of the paper. Not everyone rated all guides. It is necessary to explain better the mode of agreement and if there is an influence on the final rating.

Reviewer 5 Report

The study is well structured. The literature review is adequate, and the methodology and data analysis are as well. Also, the authors deeply discussed the results obtained and their implications. This study adds to the literature by shedding a light on the existent Mental Health and Psychosocial Support (MHPSS) Guidelines, and their methodological quality. I just have one minor suggestion: I believe it would be interesting to add some professional characteristics of the 5 independent raters (see please line 156, page 4). For instance, do all the raters have the same level of experience in the field and it was a multidisciplinary team of raters?

Reviewer 6 Report

The opening paragraph is too vague and does really lay a good foundation for the context of the study.

The goal of the study needs to be properly highlighted and justified. Instead of setting their aim in the frame of a simple question, I would recommend that the authors attempt to present the key objectives of their study with regards to what is presently known (i.e. literature), thus highlighting the added value of the article.

The authors should definitely elaborate on the hypothesis as they are not sufficiently backed with theoretical considerations.

The authors should provide the inclusion and exlusion criteria (as a paragraph or table within the methods section of the systematic. review). 

Please provide the exact keywords used in the search process.

It would be appreciated if the authors could give more details about practical implications of the study

Round 2

Reviewer 6 Report

The article is now suitale for publication